# [^68^Ga]Ga-NODAGA-E[(cRGDyK)]_2_ and [^64^Cu]Cu-DOTATATE PET Predict Improvement in Ischemic Cardiomyopathy

**DOI:** 10.3390/diagnostics13020268

**Published:** 2023-01-11

**Authors:** Bjarke Follin, Cecilie Hoeeg, Ingrid Hunter, Simon Bentsen, Morten Juhl, Jacob Kildevang Jensen, Tina Binderup, Carsten Haagen Nielsen, Rasmus Sejersten Ripa, Jens Kastrup, Annette Ekblond, Andreas Kjaer

**Affiliations:** 1Cardiology Stem Cell Centre, The Heart Centre, Copenhagen University Hospital Rigshospitalet, DK-2100 Copenhagen, Denmark; 2Department of Clinical Physiology and Nuclear Medicine & Cluster for Molecular Imaging, Copenhagen University Hospital—Rigshospitalet & Department of Biomedical Sciences, University of Copenhagen, DK-2200 Copenhagen, Denmark; 3Department of Immunology and Microbiology, University of Copenhagen, DK-2200 Copenhagen, Denmark; 4Minerva Imaging ApS, DK-3650 Oelstykke, Denmark

**Keywords:** angiogenesis, inflammation, nuclear cardiology, cardiomyopathy, cell therapy

## Abstract

An increasing number of patients are living with chronic ischemic cardiomyopathy (ICM) and/or heart failure. Treatment options and prognostic tools are lacking for many of these patients. Our aim was to investigate the prognostic value of imaging angiogenesis and macrophage activation via positron emission tomography (PET) in terms of functional improvement after cell therapy. Myocardial infarction was induced in rats. Animals were scanned with [^18^F]FDG PET and echocardiography after four weeks and randomized to allogeneic adipose tissue-derived stromal cells (ASCs, *n* = 18) or saline (*n* = 9). Angiogenesis and macrophage activation were assessed before and after treatment by [^68^Ga]Ga-RGD and [^64^Cu]Cu-DOTATATE. There was no overall effect of the treatment. Rats that improved left ventricular ejection fraction (LVEF) had higher uptake of both [^68^Ga]Ga-RGD and [^64^Cu]Cu-DOTATATE at follow-up (*p* = 0.006 and *p* = 0.008, respectively). The uptake of the two tracers correlated with each other (r = 0.683, *p* = 0.003 pre-treatment and *r* = 0.666, *p* = 0.004 post-treatment). SUVmax at follow-up could predict improvement in LVEF (*p* = 0.016 for [^68^Ga]Ga-RGD and *p* = 0.045 for [^64^Cu]Cu-DOTATATE). High uptake of [^68^Ga]Ga-RGD and [^64^Cu]Cu-DOTATATE PET after injection of ASCs or saline preceded improvement in LVEF. The use of these tracers could improve the monitoring of heart failure patients in treatment.

## 1. Introduction

Ischemic heart failure, most often caused by ischemic cardiomyopathy (ICM), impacts the lives of millions of people worldwide [1,2,3]. The prevalence of heart failure is increasing, resulting in a large group of patients with no further treatment options. This calls for better prognostic tools as well as novel treatment options [4]. 

Cell therapy is a potential new treatment for patients with ischemic heart failure, with mesenchymal stromal cells (MSCs) being the furthest in the clinical translation [5]. The result of two decades of research is moderate clinical efficacy and only a partial understanding of the MSC mode of action. Thus, additional knowledge on the mode of action and patient response is needed for further clinical development of the treatment. As with other treatments, there are responders and non-responders to therapy, indicating that there are subgroups of patients more prone to regeneration than others [6,7]. This was recently demonstrated by the results from the phase III DREAM trial, where Mesoblast publicly announced that NYHA class was important for treatment effect, though the results are not fully disclosed yet. An approach to elucidate both responders and MSC mode of action is molecular imaging of biological processes relevant to both the disease and treatment. 

Within the fields of both cardiac regenerations and cardiac patient diagnostics, angiogenesis and immune activation are key processes to assess. This can be achieved via positron emission tomography (PET). The quantification of cardiac angiogenesis through imaging of integrin α_v_β_3_ has been of particular interest. Many different tracers have been developed, but few were applied clinically [8]. Furthermore, they have predominantly been investigated in acute myocardial infarction, and not chronic ICM [9,10,11,12]. The latter is more clinically relevant for cardiac cell therapy. 

Another relevant process to quantify is macrophage activation, since macrophages are heavily involved in cardiac wound healing and remodeling [13,14]. In terms of cardiac MSC therapy, macrophage interaction has been found to be imperative for effective therapy in acute myocardial infarction and is also involved in ICM [13,15]. Imaging of the SSTR2 on activated macrophages by DOTATATE has become increasingly applied in the cardiovascular imaging [16,17].

This study aimed to investigate the value of assessing angiogenesis visualized by [^68^Ga]Ga-NODAGA-E[(cRGDyK)]_2_ (RGD) and macrophage activation visualized by the uptake of [^64^Cu]Cu-DOTA-TATE (DOTATATE) for predicting functional outcome in ICM after MSC therapy. 

## 2. Materials and Methods

### 2.1. Study Design and Animal Model

Outbred male Sprague-Dawley rats (Charles River) weighing 433 ± 19 g were included in this study. Adipose tissue-derived mesenchymal stromal cells (ASCs) were harvested from the adipose tissue of male Lewis rats. The use of ASCs from Lewis rats (Charles River) rendered the treatment allogeneic with MHC mismatch based on information from the vendor. The animals were acclimatized for two weeks prior to any operations. The animals were housed in cages with a temperature of 21 ± 2 °C and a 12:12 h dark–light cycle. During the whole experimental period, they had access to water and chow ad libitum and were continuously monitored by the on-site animal core facility. 

The study design can be seen in Figure 1, with an explanation in the figure legend. The assessment times were based on the literature and are explained in the discussion. The design was based on two ongoing phase II clinical studies using allogeneic cryopreserved ASCs in heart failure patients. Based on LVEF, the rats were randomized 2:1 to receive ASCs or saline in week 6. By treating this late after infarct induction, the treated tissue resembles the clinically relevant late-stage ischemic myocardium. The whole setup was performed in two cohorts. A total of 73 animals were operated on, while 27 were included based on having obtained the ICM phenotype characterized by low LVEF below 50% and/or clear infarct area. A total of 23 rats were scanned with [^68^Ga]Ga-RGD and 17 with [^64^Cu]Cu-DOTATATE, due to logistical challenges. 

### 2.2. Ethics

All animal experiment protocols were approved by the Danish Animal Experiments Inspectorate (Permit No. 2016-15-0201-00920). Procedures were performed in accordance with the guidelines in Directive 2010/63/EU of the European Parliament on the protection of animals used for scientific purposes. 

### 2.3. ASC Isolation and Culture

Three male Lewis rats underwent surgical resection of subcutaneous adipose tissue, which was then washed with phosphate-buffered saline (PBS) without Ca^2+^ and Mg^2+^ (Gibco). Collagenase NB 4 (Nordmark Biochemicals, Uetersen, Germany) was dissolved in Hank’s Balanced Saline Solution containing CaCl_2_ and MgCl_2_ (Gibco, Thermo Fisher Scientific, Roskilde, Denmark) in a concentration of 0.6 PZ U/mL and added to the adipose tissue 1:1. The adipose tissue was then incubated for 45 min, at 37 °C in a mini incubator with rotation (Labnet, In Vitro, Edison, NJ, USA). To isolate mononuclear cells (MNCs), the adipose tissue suspension was filtered through a 100 µm mesh (BD-falcon), centrifuged, and subsequently resuspended in Minimum Essential Medium α (αMEM) supplemented with 10% fetal bovine serum (FBS), and 100 U/mL penicillin/100 µg/mL streptomycin (P/S) (Gibco). Then, 4.5 × 10^6^ MNCs were seeded in T75 flasks (Nunc, Thermo Fisher Scientific, Roskilde, Denmark) and incubated at 37 °C, 20% O_2_, and 5% CO_2_. At approximately 90% confluency, cells were detached using TrypLE (Gibco), resuspended in CryoStor^®^ CS10 (BioLife Solutions, Bothell, WA, USA), and portioned in cryo tubes vials of 1 × 10^6^. The cells were stored in liquid N_2_. For the expansion of ASCs, P0 cells were thawed, and 1 × 10^6^ cells were seeded per T75 flask. ASCs were cultured until 90% confluency, where they were portioned in cryo tube vials of 3 × 10^6^ P1 ASCs in CryoStor^®^ and frozen until usage. 

### 2.4. ASC Phenotype

ASC phenotype was evaluated as described previously [15]. In brief, ASC phenotype was determined via flow cytometry. ASCs were thawed, washed twice in PBS, and resuspended in a concentration of 1 × 10^6^/mL in PBS. To assess ASC phenotype and quality, ASCs were labeled with Fixable Viability Stain 780 (FVS-780, BD Biosciences, Kongens Lyngby, Denmark) and subsequently washed with a fluorescence-activated cell sorter (FACS)-PBS containing 1% EDTA (Pharmacy of the Capital Region) and 10% FBS (Life Technology, Carlsbad, CA, USA) in PBS (Pharmacy of the Capital Region). Afterward, cells were stained with 5 µL antibody/100 µL cell suspension. A minimum of 2 × 10^4^ live single cells were acquired using a flow cytometer (BD FACSLyric, BD Biosciences), and data analysis was performed using FlowLogic (Inivai Technologies, Hørsholm, Denmark).

### 2.5. Infarct Induction

The infarcts were induced as described in our earlier studies [18,19]. In brief, the animals were anesthetized in 4–5% Sevoflurane, intubated, placed on a heated operation table, and ventilated (UNO micorventilator-O3, UNO Life Science Solutions, PC Zevenaar, The Netherlands). Before the operation, the animals were treated with buprenorphine 0.05 mg/kg subcutaneously, and their chests were shaven and sterilized with iodine. In sterile surgical conditions, an incision was made in the skin, with subsequent thoracotomy at the third or fourth intercostal space. The pericardium was removed gently. The left appendage was identified, and a permanent ligation was performed 3–4 mm below the appendage using a 6-0 polypropylene suture. A successful ligation would result in discoloration of the myocardium. If this was not observed, a second ligation was performed. After visual confirmation of ischemia, the ribs and skin were closed using a 4-0 vicryl suture. The animals were monitored closely in the following 72 h, with analgesic treatment by oral buprenorphine. 

### 2.6. Echocardiography

Echocardiography was performed at weeks 4 and 10 using the Vevo 3100 Preclinical Imaging Platform. The animals were anesthetized in 3% isoflurane, hair was removed from their chest, and they were placed on the accompanying plate with integrated heating, ECG, and respiratory monitoring. The peristernal long axis was identified, followed by the peristernal short axis. When a good short-axis view was established, the 4D echocardiographic acquisition was performed. This method samples images from over 60 short-axis slices to create a 3D image of the heart, complete with cardiac cycle contractions, and accounts for displacement due to respiration. The cardiac volumes are calculated from this 3D image. The echocardiography was performed by the same observer who was blinded to the treatment. 

### 2.7. Tracer Synthesis

[^68^Ga]Ga-RGD was synthesized as described earlier [10]. In brief, NODAGA-E[(cRGDyK)]_2_ (ABX GmbH, Radberg, Germany) was labeled using a Modular-Lab PharmTracer (Eckert & Ziegler). The ^68^Ge/^68^Ga generator (IGG100; Eckert & Ziegler, Berlin, Germany) was eluted with 6 mL of 0.1 HCL. The eluate was concentrated on a Strata-XC cartridge and eluted with 700 uL of NaCL/HCL. Quality control was performed using a liquid chromatograph (Ultimate 300; Dionex, MA, USA) and Kinetex C18 column (2.6 um, 100 Å, 50 × 4.6 mm; Phenomenex, Torrance, CA, USA) with UV and radio-detector connected in series. The mobile phases were eluent A, 0.1% trifluoroacetic acid in H_2_0, and eluent B, 0.1% trifluoroacetic acid in MeCN. 

[^64^Cu]Cu-DOTATATE was synthesized as described in previous studies [20]. 

2-[18F]FDG was produced as the standard formulation for clinical diagnostics. 

### 2.8. PET/CT Scan

For all PET/CT scans, the animals were anesthetized in sevoflurane, and IV access through the tail vein was established. A Siemens Inveon PET/CT scanner (Siemens, Knoxville, TN, USA) was used with a water-heated bed and monitoring of ECG and respiration. Before all PET scans, a 15-min CT scan with full rotation, 360 projections, and 65 kV was performed with 1 mL CT contrast (Omnipaque 350 mg, GE Healthcare, Denmark). 

[^18^F]-FDG was used as a control for the infarct area when quantifying [^64^Cu]Cu-DOTATATE and [^68^Ga]Ga-RGD, and to estimate the infarct area at weeks 4 and 10. The animals had been given minimal amounts of food for 5 h prior to the scans, in order to deplete the glucose in the blood. To decrease circulating free fatty acids and shift the energy consumption of the heart to glucose, 15 mg/kg Acipimox (Sigma-Aldrich, St. Louis, MO, USA) was injected subcutaneously 10 min prior to injection of 2-[^18^F]-FDG. The tail vein was cannulated using a 24 G intravenous catheter (Vasofix^®^, Braun, Denmark), and 40–45 MBq 2-[^18^F]FDG was injected. The PET scan was performed after a circulation time of 60 min. PET list-mode files were histogrammed into a single time frame. Images were reconstructed using an OSEM3D/OP-MAP algorithm with 1.5 mm resolution, 2 iterations, and 18 subsets. Images were corrected for scatter and attenuation. 

The [^64^Cu]Cu-DOTATATE tracer was supplied by Risø Campus, Danish Technical University Nanotech. The animals were scanned with [^64^Cu]Cu-DOTATATE in week 5, prior to the treatment, and week 7, the week after treatment. Then, 25–30 MBq was administered via IV to each animal with a circulation time of 90 min before PET acquisition. Images were reconstructed using an OSEM3D/OP-MAP algorithm with 3 mm resolution, 2 iterations, and 18 subsets. Images were corrected for scatter and attenuation.

The [^68^Ga]Ga-RGD tracer was supplied by the Department for Clinical Physiology, nuclear imaging, and PET at The Copenhagen University Hospital Rigshospitalet. The animals were scanned with [^68^Ga]Ga-RGD in week 5 and two weeks after treatment in week 8. The same volume was injected via IV into each animal to keep the amount of peptide constant, and the PET acquisition was performed after a circulation time of 30 min. The differences in injected activity were compensated with differing acquisition times in such a way that the PET data were acquired for 10 min with animals receiving >35 MBq, 15 min with 20–35 MBq, and 20 min with <20 MBq. Images were reconstructed via an OSEM3D/OP-MAP algorithm with 2 mm resolution, 2 iterations, and 18 subsets. Images were corrected for scatter and attenuation.

### 2.9. Image Analysis

Inveon Acquisition Workspace was used to reconstruct PET and CT images, and Inveon Research Workspace (both Siemens, Knoxville, TN, USA) was used for co-registration of all PET and CT images and for ROI creation and data collection for both [^64^Cu]Cu-DOTATATE and [^68^Ga]Ga-RGD. ROIs were drawn simultaneously on images from the 5th-week scan and the follow-up from either week 7 or 8. The images 2-[^18^F]-FDG from weeks 4 and 10 guided the outline of the infarct area. 

The static 2-[^18^F]FDG images were analyzed using Corridor4DM version 2017 (Invia LLC, Ann Arbor, MI, USA). The images were reoriented into a short axis, a vertical long axis, and a horizontal long axis. The myocardial contour was automatically outlined and subsequently inspected manually by experienced professionals. The images were excluded for further analyses if the quality was too low for proper automatic recognition. The uptake was compared to a normal database created from other male Sprague–Dawley rats. The infarcted area was automatically calculated as a summed score of metabolic defects in the American Heart Association (AHA) 17-segment model. 

### 2.10. Treatment

Treatment was performed using the Vevo 3100 Preclinical Imaging Platform. The animals were anesthetized in 3–4% Isoflurane, and hair was removed, as mentioned previously, in addition to receiving analgesic treatment with buprenorphine subcutaneously prior to ASC treatment. The cryopreserved ASCs were centrifuged, the supernatant was removed, and the cells were resuspended to reach a concentration of 10 × 10^6^ ASCs pr. ml CryoStor10 in the syringe. The injection needle was aligned with the transducer, after which the heart was localized in a short-axis view. The needle was inserted in the myocardium through the thorax, with needle penetration of the myocardium being confirmed by 2–3 arrhythmic ECG peaks. The injection was performed and visualized as bleaching of the myocardium on the echo image. Injections were performed anterior and lateral/inferior to the myocardial infarction (MI). The needle was retracted, and the animals were observed after the procedure. The operator was blinded to which animals were treated with ASCs or saline. 

### 2.11. Immunohistochemistry

After the last echocardiographic scans at week 10, the animals were anesthetized and euthanized by decapitation. The hearts were quickly excised, and retrograde perfusion was performed through the aorta first using saline, followed by paraformaldehyde. The hearts were subsequently fixated in 4% paraformaldehyde for 24 h before being embedded in paraffin. 

Four µm short axis slices were prepared using a Microm HM 355S microtome (Thermo Scientific) and mounted on charged microscope slides (SuperFrost, Thermo Scientific). The slices were stained for myocardial differentiation using Hematoxylin and Eosin (HE) and collagen deposition with Masson’s Trichrome (MT). None of the antibodies chosen for integrin α_v_β_3_ and SSTR2 did not demonstrate proper specificity. Therefore, we chose CD31 and CD68 for immunohistochemistry instead. These are not the actual targets of the tracers but have previously demonstrated correlation with the tracer uptake [12,20]. As such, immunohistochemical staining for endothelial cells (CD31, 1:50, Novus Bio) and macrophages (CD68, 1:3000, Abcam) was performed.

All slices were scanned with an Axio scan Z1 (Zeiss, Köln, Germany) and then examined using Image J (Fiji). The fibrotic area was quantified via a previously applied macro for Image J. 

### 2.12. Statistics

The normality of data was tested with Kolmogorov–Smirnov and Shapiro–Wilk tests, while Levene’s test was used to assess equal variance. Differences in functional parameters and tracer uptake between groups were assessed via a student’s *t*-test or the non-parametric Wilcoxon signed rank test. Person’s correlation was performed when data had Gaussian distribution, while Spearman’s correlation was performed if this was not the case. Comparisons of pre- and post-treatment measures were performed via a paired samples *t*-test. All statistics were performed in SPSS (IBM SPSS version 25), and all graphs were created with the GraphPad Prism (9.3.1). 

## 3. Results

### 3.1. Rat ASC Characterization and Injection

The results from the characterization of the allogeneic rat ASC product have been described previously [15]. In brief, mesenchymal markers CD29, CD73, and CD90 were expressed at levels above 95%, while endothelial marker CD31, hematopoietic marker CD45, and monocyte/macrophage marker CD11b/c were all lower than 2%. 

The intra-myocardial echo-guided trans-thoracic injection method was verified previously by injecting Luc2_Tdtomato-transduced human ASCs in a healthy rat heart, as well as by tracking Y-chromosome DNA by qPCR [15]. 

### 3.2. Tracer Uptake in the Infarct Area

Infarct-specific tracer uptakes were detected for both the [^64^Cu]Cu-DOTATATE and [^68^Ga]Ga-RGD tracers (Figure 2A). The background was defined as the inferior left ventricular wall. When including all animals across treatment groups, the SUV_max_ for [^68^Ga]Ga-RGD in the infarct area was 0.88 ± 0.22 at week five and 0.80 ± 0.16 at week eight (background of 0.62 ± 0.12 and 0.63 ± 0.11, respectively) (See Figure 2B). The target-to-background ratio (TBR) decreased significantly from 1.45 ± 0.14 to 1.29 ± 0.13 (*p* < 0.001). TBR was only included to demonstrate infarct-specific uptake. SUV_max_ was used for the remainder of the article. 

The SUV_max_ for the [^64^Cu]Cu-DOTATATE tracer in the infarct area was 0.42 ± 0.08 at week five and 0.45 ± 0.13 at week seven (background of 0.34 ± 0.04 and 0.36 ± 0.09, respectively). See Figure 2C. The TBR was 1.23 ± 0.13 at week five and 1.26 ± 0.13 at week seven. 

There was a significant correlation between the uptake of [^64^Cu]Cu-DOTATATE and [^68^Ga]Ga-RGD in the infarct area. This was significant both at baseline in week five (r = 0.683, *p* = 0.003) and follow-up in weeks seven and eight, respectively, (r = 0.663, *p* = 0.004). See Figure 2D. 

### 3.3. Effect of ASC Treatment

The rats in the ASC group weighed significantly more than the rats in the saline group at week 10 (*p* = 0.015, Figure 3B). There was a tendency for the same difference to be present at week four. 

There was no difference in the cardiac function measured by left ventricular ejection fraction (LVEF) between the saline and ASC groups (from 48.4 ± 9.3 to 48.5 ± 12.0 in the ASC group, and from 48.4 ± 9.1 to 48.8 ± 10.2 in the saline group, *p* = 0.860) (Figure 3C). In addition, there was no difference in left ventricular end-systolic volume, end-diastolic volume, or infarct sizes across groups. The infarct sizes measured by 2-[^18^F]FDG uptake defects had a significant negative correlation with LVEF (r = −0.625, *p* < 0.0001). See Appendix A.

There were no significant differences in the uptake of [^68^Ga]Ga-RGD and [^64^Cu]Cu-DOTATATE in the infarct area between saline and ASC groups (See Figure 3D,E). Specific values are presented in Table 1.

### 3.4. Increased RGD Uptake in Hearts with Improved Pump Function

We observed a significant correlation (r = 0.459, *p* = 0.028) between the change in LVEF and SUV_max_ of [^68^Ga]Ga-RGD in the infarct area two weeks after the treatment (Figure 4A). To investigate the relationship between tracer uptake and functional recovery, the rats were divided into two groups based on ΔLVEF. Based on the available literature on the prediction of responders and non-responders to cardiac cell therapy, we chose a cut-off of ΔLVEF > 5 percentage points for grouping rats as “Improved” (*n* = 7; 5 from ASC and 2 from saline) as opposed to “Not improved” (*n* = 16; 9 from ASC and 7 from saline) [6,7]. It was clear that the uptake was different between groups. The uptake of [^68^Ga]Ga-RGD decreased over time from 0.93 ± 0.13 to 0.75 ± 0.14 in the rats, which did not improve their LVEF, while the uptake did not drop in the improved group (from 0.81 ± 0.29 to 0.91 ± 0.17, Figure 4C). The SUV_max_ for the rats in the improved group was significantly higher than for the others (*p* = 0.006). There was a tendency for the difference between groups in the remote area of the heart, with an increase in uptake in the improved rats, only this was not significant (Appendix A, *p* = 0.061). Finally, we found that the SUV_max_ in the infarct area two weeks after treatment could predict improvement in LVEF via a receiver operating characteristic (ROC) analysis (Figure 4D, the area under the curve = 0.82, *p* = 0.016, sensitivity: 71% specificity: 88%).

### 3.5. Increased [^64^Cu]Cu-DOTATATE Uptake in Hearts with Improved Pump Function

We did not find the same correlation between the change in LVEF and SUV_max_ of [^64^Cu]Cu-DOTATATE in the infarct area one week after treatment (Appendix A). The rats were divided into improved (*n* = 5; 3 from ASC and 2 from saline) and not improved (*n* = 12; 7 from ASC and 5 from saline) groups. The uptake in the improved rats increased from 0.40 ± 0.08 to 0.56 ± 0.14, resulting in significantly higher SUV_max_ values one week after the treatment compared to the uptake in the group that did not improve (from 0.43 ± 0.08 to 0.41 ± 0.09, Figure 5A, *p* = 0.008). As with [^68^Ga]Ga-RGD, the same tendency of difference between groups was observed in the remote area of the heart, though not as significant (Appendix A, *p* = 0.084). Furthermore, we found that the [^64^Cu]Cu-DOTATATE uptake one week after the treatment could predict functional improvement with a ROC test (Figure 5B, the area under the curve = 0.82, *p* = 0.045, sensitivity: 80% specificity: 83%). 

### 3.6. Immunohistochemistry

The fibrotic area visualized by Masson’s Trichrome correlated significantly with the infarct size measured by 2-[^18^F]FDG (Appendix A, r = 0.550, *p* = 0.024). There were no correlations between tracer uptake and CD68-positive area or the number of counted CD31-positive vessels. There were no differences between the groups that improved and the groups that did not improve on Masson’s Trichrome (*p* = 0.605), CD68 (*p* = 0.428), or CD31 (*p* = 0.210). 

## 4. Discussion

The results suggest that the ASC treatment does not lead to a functional effect but that both tracers were predictive for functional outcomes. This indicates that imaging the biological processes associated with integrin α_v_β_3_ and SSTR2 expression could lead to improved monitoring of heart failure patients in treatment. 

We have previously shown infarct-specific [^68^Ga]Ga-RGD-uptake in a pig and rat model of acute myocardial infarction [10]. This is the first study investigating the uptake five and eight weeks after MI. The overall uptake follows the same tendency as our previous studies, that the infarct-specific [^68^Ga]Ga-RGD-uptake decreases over time. This is expected as the degree of angiogenesis decreases as the myocardial wound contracts. The fact that the [^68^Ga]Ga-RGD uptake in the improved group increased in both the infarct and non-infarct areas suggests that the positive remodeling responsible for the functional improvement is not restricted to the infarct area. 

In our study, the uptake of [^68^Ga]Ga-RGD did not correlate to LVEF at the same time point. However, change in LVEF and post-treatment uptake correlated. This demonstrates that the [^68^Ga]Ga-RGD tracer detects a perfusion defect and instead is related to function in ICM. This is in line with Jenkins et al., who found that the uptake of their integrin α_v_β_3_-specific tracer, [^18^F]-Fluciclatide, was increased in segments with functional improvement after acute myocardial infarction [21]. Likewise, the ischemic heart failure patients treated with autologous cardiac MSCs tended to result in higher uptake of [^99m^Tc]-NC100692 targeting integrin α_v_β_3_ at four days after the treatment, as well as improve LVEF [22]. This was a feasibility study and not powered to show significance in terms of treatment effect. 

Most integrin α_v_β_3_ tracers have been investigated in acute myocardial infarction models and patients, and thus, the results should be compared with caution. In a clinical study, subacute myocardial infarction (31 days after injury) displayed uptake of an integrin α_v_β_3_ PET tracer, and the uptake correlated with infarct size and perfusion defect [23]. In animal studies, one group found that cell therapy with a 3D aggregate of MSCs and endothelial cells increased integrin α_v_β_3_ detection five days after treatment and significantly decreased infarct size at four weeks [24]. Another group found that therapy with cardiac-induced cells resulted in lower integrin α_v_β_3_ detection seven days after treatment but improved function [25]. This demonstrates that both integrin α_v_β_3_ tracers and different cell therapies are not interchangeable and that these studies must be performed for different combinations of tracers and cell therapies. Both studies were compared on a group basis, with no correlation or prediction analyses, making it difficult to decipher the value of the imaging on an individual level. 

The present study is the first to apply DOTATATE imaging to assess the cardiac tissue before and after cell therapy and the first animal study to investigate DOTATATE in a non-acute ischemic condition. The association between [^64^Cu]Cu-DOTATATE and pump function was not as strong as for [^68^Ga]Ga-RGD. This could be due to the biological processes detected by the two tracers or the lower number of animals imaged with the [^64^Cu]Cu-DOTATATE tracer due to logistic challenges. 

The uptake of [^64^Cu]Cu-DOTATATE and [^68^Ga]Ga-RGD were correlated both pre- and post-treatment and, furthermore, related to functional improvement. The functional relation suggests that the image-activated macrophages are a part of wound healing and may be more anti-inflammatory than pro-inflammatory [20,26]. Integrin α_v_β_3_ is expressed on activated endothelial cells but also on activated macrophages and myofibroblasts and generally appears central in the coordination of the myocardial repair [21]. As such, some of the PET signals from the RGD uptake may be contributed by the same cells contributing to the DOTATATE signal. However, the significant decrease in TBR of [^68^Ga]Ga-RGD over time in contrast with the more stable [^64^Cu]Cu-DOTATATE suggests different tissue processes and that [^68^Ga]Ga-RGD images are more than inflammation. Furthermore, this suggests that macrophage activation is more stable than the remaining wound healing process at five–eight weeks after MI, which is consistent with the literature [27]. 

Our results on the lack of functional benefit in the chronic model are on par with those published by Vagnozzi et al., who demonstrated that cell therapy via mononuclear bone marrow cells was effective in an acute animal model but had no effect in a chronic model [13,28]. The administration of saline or ASC induces a micro-trauma in the myocardium, resulting in a regular wound healing cascade. Our hypothesis was, that this micro-trauma combined with ASC presence would result in a higher degree of macrophage activation and angiogenesis in the ASC group, leading to improved cardiac function. The results suggest that the micro-trauma itself is more important than the administration. 

Assessment of macrophage activation one week after treatment was based on previous experience, with this being the earliest time point to identify potential differences between saline and ASC treatment[15]. In addition, it is established that reparative macrophages peak seven days after cardiac injury in rodents [27,29]. Expression of integrin α_v_β_3_ after a cardiac injury has been demonstrated to peak between week 0 and week 3, and, therefore, we chose to assess angiogenesis two weeks after treatment [30]. 

### Limitations

There was a tendency for the weight to be higher in the ASC group at baseline, which may explain the significant difference in weight between the treatment groups at week 10. The other results do not indicate that the bias in weight influenced the other parameters.

We did not include a sham group in the study since the infarct specificity of both tracers has already been demonstrated in other studies [10,31]. 

There is a risk of inflated alpha values since the study was not originally designed for comparison of improved and not improved rats across groups. 

The direct correlation between immunohistochemistry and tracer uptake was hampered by available antibodies and the study design. The cardiac tissue was harvested for immunohistochemistry after the echocardiography at week 10, while the tracer uptakes were measured at week 7 and week 8. As such, care should be taken when drawing conclusions based on the correlation analyses between immunohistochemistry and tracer uptake. 

## 5. Conclusions

This Cardiac uptake of [^68^Ga]Ga-RGD and [^64^Cu]Cu-DOTATATE correlated, and the high uptake predicted functional improvement in chronic ICM rats. The uptake was increased in both the infarcted and non-infarcted areas in the improved rats. There was no functional benefit of treatment with cryopreserved allogeneic ASCs. Our results suggest that cardiac imaging using [^68^Ga]Ga-RGD and [^64^Cu]Cu-DOTATATE could potentially be used to assess beneficial tissue processes and improve prognosis after therapy in ICM and potentially heart failure in patients. This will need to be explored in more extensive studies. 

## Figures and Tables

**Figure 1 diagnostics-13-00268-f001:**
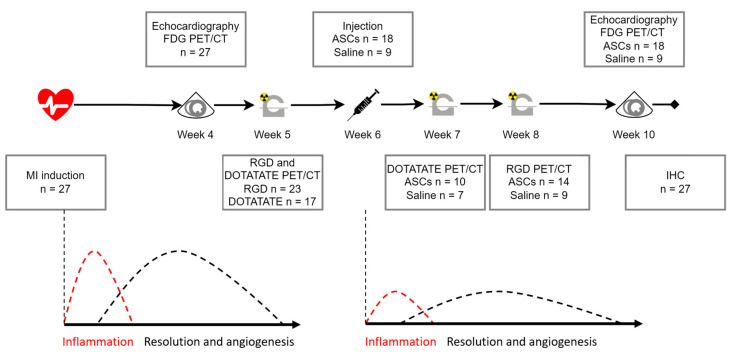
Study design with hypothesized cardiac wound healing phases. MI was induced at week 0, and the animals were allowed to recover for 4 weeks, to ensure the acute effect of the infarct was diminished. Echocardiography and 2-[^18^F]FDG PET/CT for cardiac volumes and infarct area, respectively, were performed at week 4 for randomization. Pre-treatment PET/CT scans via [^64^Cu]Cu-DOTATATE to detect baseline inflammation and [^68^Ga]Ga-RGD to detect baseline angiogenesis was performed at week 5. In week 6, animals were treated with a trans-thoracic intra-myocardial injection of cryopreserved allogeneic rat ASCs or saline. The design was based on the hypothesis that treatment would initiate a new round of wound healing. Post-treatment scans with [^64^Cu]Cu-DOTATATE were performed in week 6 (one week after treatment), while [^68^Ga]Ga-RGD scans were performed in week 8 (two weeks after treatment). After follow-up echocardiography and 2-[^18^F]FDG PET/CT in week 10 (four weeks after treatment), the rats were euthanized in week 11. MI: myocardial infarction; ASCs: Adipose tissue-derived stromal cells; PET/CT: positron emission tomography/computed tomography. The wound healing hypothesized time was based on Prahbu et al. as well as unpublished data.

**Figure 2 diagnostics-13-00268-f002:**
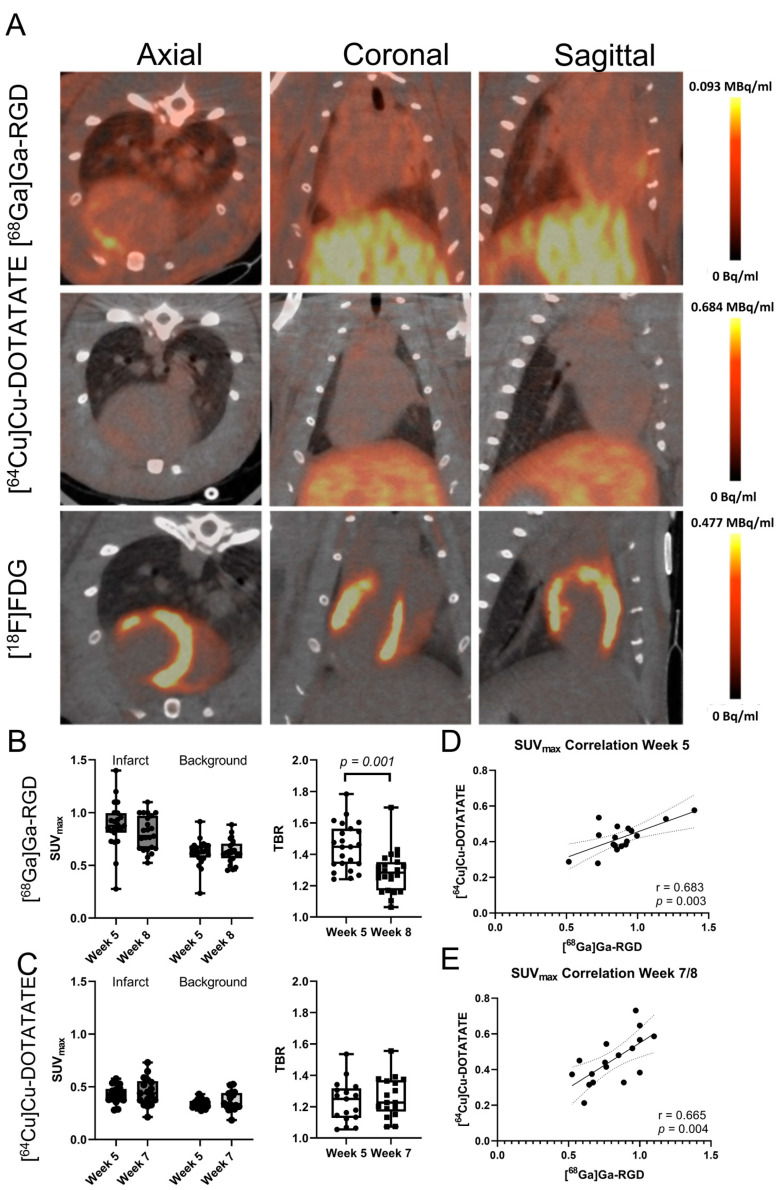
Uptake of [^68^Ga]Ga-RGD and [^64^Cu]Cu-DOTATATE in the same rat. The tracers showed infarct-specific uptake in the anterolateral part of the heart (arrows), in the same area which did not have an uptake of 2-[^18^F]FDG (**A**). Uptake of [^68^Ga]Ga-RGD in terms of SUV_max_ and TBR (**B**), *n* = 23. Uptake of [^64^Cu]Cu-DOTATATE in terms of SUV_max_ and TBR (**C**), *n* = 17. Correlation between the uptake of [^68^Ga]Ga-RGD and [^64^Cu]Cu-DOTATATE before (**D**) and after treatment (**E**). SUV: Standard uptake value; TBR: Target-to-background ratio.

**Figure 3 diagnostics-13-00268-f003:**
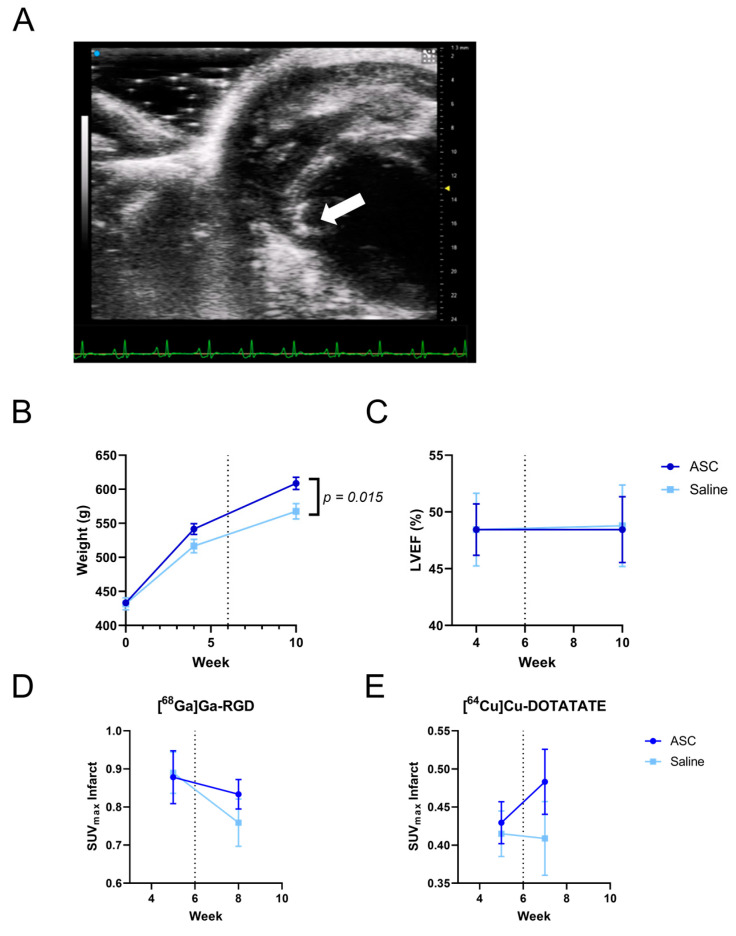
Treatment effect of ASCs. Trans-thoracic intra-myocardial injection (**A**). The needle is visible outside the chest, and the injected fluid can be seen as small white bubbles in the myocardium (arrow). There was a significant difference in the weight of the animals between groups (**B**) but no significant effect on LVEF (**C**) or uptake of [^68^Ga]Ga-RGD (**D**) and [^64^Cu]Cu-DOTATATE (**E**), n = 27. LVEF; left ventricular ejection fraction.

**Figure 4 diagnostics-13-00268-f004:**
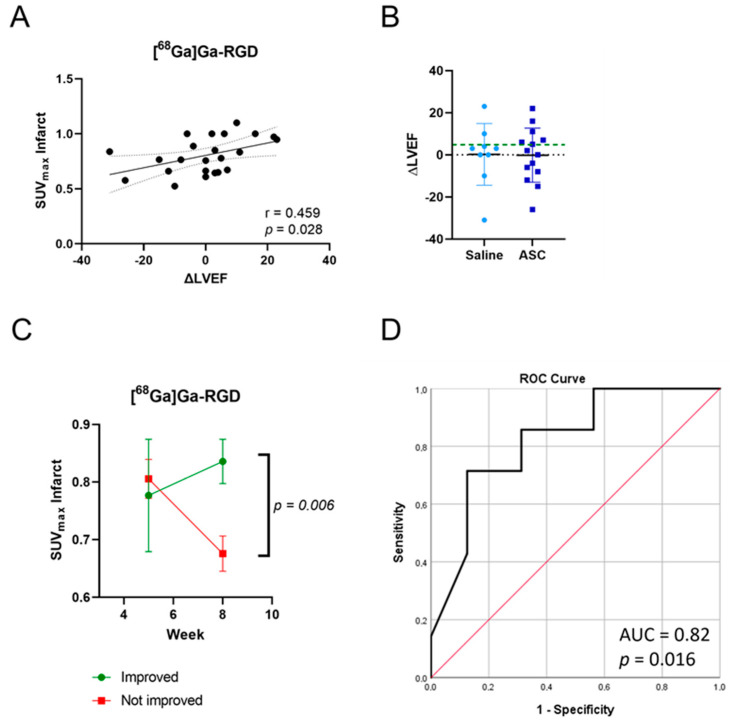
Correlation between [^68^Ga]Ga-RGD uptake and change in LVEF (**A**). Animals were grouped based on their changes in LVEF (**B**), with 5 percentage points being the cut-off between groups (the green dotted line). Differences in uptake of [^68^Ga]Ga-RGD between animals that improved LVEF and those that did not (**C**). ROC analysis for predicting LVEF based on [^68^Ga]Ga-RGD uptake with SUV_max_ at week 8 (two weeks after treatment) (**D**), *n* = 23. ROC: Receiver-operating-characteristic.

**Figure 5 diagnostics-13-00268-f005:**
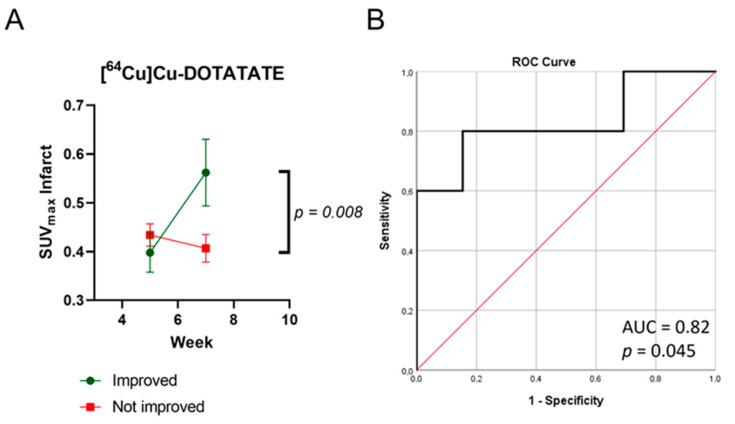
Differences in uptake of [^64^Cu]Cu-DOTATATE between animals that improved LVEF and those that did not (**A**). ROC analysis for predicting LVEF based on [^64^Cu]Cu-DOTATATE uptake with SUV_max_ at week 7 (one week after treatment) (**B**), *n* = 17.

**Table 1 diagnostics-13-00268-t001:** Cardiac volumes and tracer uptake across groups.

	**ASC (n = 18)**		**Saline (n = 9)**	
	**Pre-Treatment**	**Post-Treatment**	**Pre-Treatment**	**Post-Treatment**
LVEF (%)	48.4 ± 9.3	48.5 ± 12.0	48.4 ± 9.1	48.8 ± 10.2
LVESV (µL)	296.9 ± 119.8	294.4 ± 88.8	289.2 ± 78.6	249.5 ± 109.5
LVEDV (µL)	573.5 ± 166.4	582.6 ± 163.1	561.0 ± 103.5	626.7 ± 377.4
[^68^Ga]Ga-RGD SUV_max_	0.88 ± 0.25	0.83 ± 0.14	0.89 ± 0.15	0.76 ± 0.18
[^64^Cu]Cu-DOTATATE SUV_max_	0.43 ± 0.08	0.48 ± 0.13	0.41 ± 0.07	0.41 ± 0.12
	**Improved (n = 8)**		**Not Improved (n = 19)**	
	**Pre-Treatment**	**Post-Treatment**	**Pre-Treatment**	**Post-Treatment**
LVEF (%)	42.4 ± 9.5	55.1 ± 7.3	51.0 ± 7.8	45.8 ± 11.7
LVESV (µL)	353.2 ± 157.0	270.4 ± 113.1	269.5 ± 63.9	283.2 ± 91.4
LVEDV (µL)	598.2 ± 196.9	597.1 ± 181.5	557.2 ± 120.5	597.4 ± 281.8
[^68^Ga]Ga-RGD SUV_max_	0.81 ± 0.29	0.91 ± 0.17	0.93 ± 0.13	0.75 ± 0.14
[^64^Cu]Cu-DOTATATE SUV_max_	0.40 ± 0.08	0.56 ± 0.14	0.43 ± 0.08	0.41 ± 0.09

Pre-treatment time is at week 4 for cardiac volumes and week 5 for tracer uptake. Post-treatment time is week 10 for cardiac volumes, week 7 for [^64^Cu]Cu-DOTATATE, and week 8 for [^68^Ga]Ga-RGD. Number of animals varies between measured parameters. See the main text for specific numbers.

## Data Availability

The data supporting the findings of this study are available within the article.

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
