# Peer review of "[68Ga]Ga-NODAGA-E[(cRGDyK)]2 and [64Cu]Cu-DOTATATE PET Predict Improvement in Ischemic Cardiomyopathy"

_diagnostics, 2023, doi:10.3390/diagnostics13020268_

Round 1
Reviewer 1 Report
It is a heavy animal study with many interventions (induction of myocardial infarction, multiple imaging at several follow-up times, cell therapy).
Unfortunately the data on cell therapy are negative. This is why the data on imaging should be put forward in the discussion and the possible perspectives in humans must be developed (monitoring of response for remodeling, proposal of other therapies than cell therapy).
Be careful, there is a risk of inflation of the alpha risk because the statistics comparing the groups improving their ejection fraction were probably not foreseen at the beginning of the study. This must be included in the limits.
Author Response
We thank the reviewer for the good comments.
We agree that the imaging data should be the focus of the discussion. We have rearranged the discussion, so that the discussion regarding treatment and micro trauma is at the end of the discussion. This means that the discussion regarding imaging is emphasized. We have also added the following text in our conclusions to emphasize the perspectives of the imaging results:
“Our results suggest that cardiac imaging using [68Ga]Ga-RGD and [64Cu]Cu-DOTATATE could potentially be used to assess beneficial tissue processes after therapy in ICM and potentially heart failure patients. This will need to be explored more extensive studies.”
Potential inflation of the alpha value is a concern, and we have added the following to the limitations:
“There is a risk of inflated alpha values since the study was not originally designed for comparison of improved and not improved rats across groups.”
We hope that the changes are sufficient.
Reviewer 2 Report
The authors Follin et al. present a very interesting study tesing the use of [68Ga]Ga-RGD and of [64Cu]Cu-DOTATATE for marker of chronic ischemic cardiomyopathy development after myocardial infarction. The methodology is very well descibed. All methods are described and used adequately. The authors report the results in a well written section. the discussion is to the point. The only comment addresses the paragraph in the results section about immunohistochemistry. This paragraph should be expanded to include a comparison of the results in hearts with improved vs. without improved pump function.
Author Response
We thank the reviewer for the comments. We agree, and have expanded the paragraph to include the following text:
“There were no differences between the improved and not improved groups on Masson’s Trichrome (p = 0.605), CD68 (p = 0.428), or CD31 (p = 0.210).”
We hope that the changes are sufficient.